# Neurodevelopment of Autism: Critical Periods, Stress and Nutrition

**DOI:** 10.3390/cells13231968

**Published:** 2024-11-28

**Authors:** George Ayoub

**Affiliations:** Psychology, Santa Barbara City College, Santa Barbara, CA 93109, USA; ayoub@psych.ucsb.edu

**Keywords:** ASD, folate, taurine, cysteine, autism, prenatal development, critical period

## Abstract

Autism spectrum disorder (ASD) is a neurodevelopmental disability that presents significant challenges in communication and behavior. ASD prevalence exceeds 2% among eight-year-old children and is at similar levels globally. We propose that critical periods during fetal development and early postnatal years establish the conditions for either neurotypical development or the emergence of autism through mechanisms that influence immune function or delay neuronal development. One critical period is characterized by the requirement for folate, a crucial methyl donor needed for DNA regulation. Insufficient folate availability has been linked to the risk of developing ASD. Another critical period may be affected by oxidative stress or inflammation of the fetal brain, potentially due to inadequate microglial immunity, which can lead to CNS inflammatory changes that disrupt typical neurodevelopment. We suggest that early supplementation with reduced folate and taurine during both the fetal and postnatal stages may be effective in mitigating the severity of ASD symptoms by promoting neurotypical development through these critical neurodevelopmental periods.

## 1. Introduction

Autism spectrum disorder (ASD) is a neurodevelopmental condition characterized by a range of variants that exhibit similar symptoms. One notable variant is cerebral folate deficiency (CFD), also called folate anemia, which is estimated to affect up to 70% of ASD children [1,2]. This condition is linked to the presence of an autoantibody targeting folate receptor alpha, the primary transporter of folate from the bloodstream to the brain and from mother to fetus. Research indicates that when dairy products are eliminated from the diet of affected children, levels of this autoantibody decrease, leading to a restoration of folate levels in the cerebrospinal fluid [3].

Emerging evidence suggests that maternal infections, exogenous agents, or traumatic events may also contribute to the risk of ASD. Inflammatory responses during pregnancy can negatively impact fetal development, increasing the likelihood of autism [4,5,6]. Additionally, fecal transplants have demonstrated effectiveness in improving communication symptoms in autistic children, highlighting the microbiome’s crucial role in metabolizing nutrients vital for typical neurodevelopment [7].

Taurine, an amino acid, has been identified as another factor that may play a role, with lower taurine levels correlating with an increased prevalence of autism symptoms [8,9]. Our hypothesis posits that folate and taurine each have critical roles during essential developmental periods, possibly by reducing cerebral inflammation, and their depletion may further heighten the risk of developing autism. Although the specific bodily conditions leading to each ASD variant remain unclear, we suggest that these biomarkers may also interact synergistically, influencing the severity of ASD symptoms. This area warrants further investigation.

## 2. Development

### 2.1. Folate

Folate (vitamin B-9) is essential for neurodevelopment, serving as a methyl donor crucial for DNA regulation. While folate is naturally present in various foods, dietary supplements typically contain oxidized folic acid. Each individual can convert up to 0.5 milligrams of folic acid into methyl folate daily.

The presence of folate receptor autoantibody (FRAA) inhibits folate receptor alpha, a high-affinity transporter vital for transferring folate across the blood–brain barrier [10,11]. Clinical studies indicate that dietary changes, particularly the removal of dairy products, can lower blood levels of FRAA [3]. This decrease may facilitate the restoration of folate accumulation in the brain. Although FRAA is found in less than 10% of the general population, approximately 70% of autism spectrum disorder (ASD) children are positive for FRAA [2]. Meta-analyses indicate that ASD children are 20 times more likely to have FRAA, suggesting a genetic predisposition that impairs transfer of vitamin B-9 across the blood–brain barrier [11].

Despite the potential to reduce FRAA levels through significant dietary adjustments [3], many ASD children exhibit resistance to such changes. This resistance contributes to the ongoing dietary factors that elevate FRAA levels, resulting in cerebral folate deficiency (CFD) due to diminished folate in the brain. Treatment for this deficiency often involves increasing the levels of reduced folate, such as methyl-folate or folinic acid [10,11,12]. Because natural forms of folate (which is folate in reduced form) can cross the blood–brain barrier via the low-affinity transport mechanism unaffected by FRAA, achieving adequate cerebral folate levels necessitates higher concentrations in the bloodstream.

While antibodies are generally unable to penetrate the adult brain due to the blood–brain barrier, FRAA effectively prevents folate transfer into the brain. Furthermore, autoantibodies produced during pregnancy cross the placenta and can target fetal brain proteins, a phenomenon linked to maternal autoantibody-related autism, which is associated with a diagnosis of ASD [4].

### 2.2. Immune

Research has identified specific autoantibody targets commonly found in mothers of children later diagnosed as on the autism spectrum. The presence of these circulating maternal autoantibodies correlates with an increased incidence of stereotypical behaviors in affected children. Maternal stressors, including infections and exposure to environmental toxins, can activate the immune response, leading to the production of autoantibodies that interact with fetal brain proteins [13,14]. This immune response has been implicated in the pathogenesis of neuropsychiatric disorders, including ASD [5].

Both human and animal studies have established a connection between maternal stress and development of ASD. Maternal immune activation triggered by viral or bacterial infections during pregnancy, along with exposure to environmental toxins, has been associated with a heightened risk of ASD in offspring [4]. In this regard, maternal stress appears to play a pivotal role in initiating the immune response that may induce neurodevelopmental changes, contributing to the onset of neuropsychiatric disorders such as ASD [13,15,16].

### 2.3. Taurine

Taurine is an amino acid that plays a significant non-protein role in neurodevelopment, immunity, and as an antioxidant. It is found in higher concentrations in the brain compared to blood plasma, with the highest levels observed during the fetal period.

Studies have highlighted taurine’s essential functions as an antioxidant, where it mitigates oxidative stress impacting mitochondrial enzymes, and as an anti-inflammatory agent by inhibiting cytokine production. Additionally, taurine serves as a neuroprotectant, particularly in the context of epilepsy, where it reduces seizure activity by binding to GABA_A_ receptors [8].

The anti-inflammatory effects of taurine are mediated through the reduction of microglial activation [17]. Microglia play a crucial immune role within the central nervous system, and in ASD their overactivation can hinder synaptic pruning, leading to an increase in excessive dendritic spines in the hippocampus. Studies involving fetal neuroprogenitor cells have shown that taurine enhances cell proliferation and the development of neurons. In mouse models, taurine administration has been associated with increased neuronal count and synaptogenesis [18].

Neurogenesis and synaptic pruning are critical processes occurring during interconnected neurodevelopmental periods. A reduction in neurogenesis and subsequent impairments in synaptic pruning during these critical windows can have lasting effects, potentially predisposing individuals to later ASD. While some researchers have proposed taurine as a potential biomarker for ASD [19], findings regarding plasma taurine levels remain inconclusive, with studies reporting only minor differences between autistic children and their neurotypical siblings [9,20].

Taurine is synthesized from cysteine through oxidative processes. Given that plasma cysteine levels are notably low in children diagnosed as ASD [21,22], the oxidative stress pathway associated with autism may implicate both cysteine and taurine. Supplementation with N-acetylcysteine, which is converted into glutathione and cysteine within the body, has been explored in a clinical trial and was found to reduce hyperactivity in these children [23]. While the literature reveals that the modification of inflammation does not have a major clinical effect, we postulate that timing is essential. Since taurine is synthesized in humans and is gained through the consumption of animal products, we suggest that inflammation may reduce production and that this reduction, if near a critical period, may have lasting effect. We suggest that this may be assessed in a retrospective study comparing outcomes of those with an animal product diet and those who are vegan/vegetarian to determine if inflammation during pregnancy (for example, due to infection) has a more long-term impact on the child’s neurological development.

Moreover, lower plasma cysteine levels in autistic children appear to correlate with elevated plasma homocysteine concentrations [24]. This increase in homocysteine may stem from diminished vitamin B levels, which could further contribute to cerebral folate deficiency. Since vitamin B9 in its reduced form lowers homocysteine [25], treatment with reduced folate may be impactful for both lowering inflammation and by reducing CFD.

### 2.4. Clinical Trial Support

Multiple clinical trials indicate that autistic children with folate receptor autoantibodies (FRAAs) may experience improvement in communication abilities when supplemented daily with reduced folate, such as folinic acid, over a three-month period [10,25,26,27]. The results consistently show about two thirds of children receiving folinic acid supplementation show improvement in their communication. This suggests that reduced folate supplementation can effectively address cerebral folate deficiency (CFD) in a majority of affected children. When CFD is mitigated or resolved, it may then be feasible to adjust the diets of ASD children to lessen or potentially eliminate production of FRAA, thereby alleviating ASD symptoms. To date, the trials have been of modest size (approx. 20 children in each) and limited to three months in duration. Each has found that of all the ASD children in the trial, 70% had FRAA and of these FRAA positive children, two thirds responded to the folinic acid with improvement in their communication.

Key components of such a dietary intervention involve eliminating foods that may trigger FRAA production while incorporating natural folate sources. A diet rich in natural folate—similar to the Mediterranean diet, which includes fresh fruits and vegetables, nuts, legumes, fish, and olive oil—may prove beneficial.

Additionally, one report indicates that a fecal transplant to an ASD child from a neurotypical sibling may also address CFD and improve communication symptoms [28]. This may be an alternate means to repair CFD, but folate supplementation followed by dietary changes currently has stronger support.

The value of these trials has been to show that FRAA is present in a majority of ASD children (but only in about 5% of non-ASD children) and that reduced folate can mitigate at least some of the deficits, indicating folate is a key player. There is need for a larger trial that would bring this work to the fore as a potential best practice and reveal any categories of greater effectiveness. There is also a need to have data that extend past the three months to identify if there may be further advances seen.

However, it is important to recognize that the contribution of FRAA to CFD begins early in development, particularly during critical neurodevelopmental periods. Research has demonstrated that when either biological parent presents with FRAA, their child faces an increased risk of developing ASD. Elevated prenatal FRAA, especially when it is present in the mother or father, is associated with a heightened risk of childhood ASD. One study revealed that 75.6% of autistic children exhibited FRAA, while the prevalence among their mothers was 34% and among their fathers was 29%, compared to only 3% in healthy controls. Another investigation found FRAA prevalence rates of 76% in autistic children, 75% in unaffected siblings, 69% in fathers, and 59% in mothers, while the prevalence in unrelated normal controls was only 29% [27,29,30]. The similar prevalence in fathers and mothers may point to a genetic origin for FRAA, with perhaps a maternal response to the fetus or placenta triggering FRAA production.

Similarly, a new report indicates that treatment of FRAA-positive children with folinic acid is most effective in improving communication if the child is 2–3 years of age, with efficacy dropping by 50% for 5-year-old children and having a minimal impact for children over 7 years of age [31].

### 2.5. Critical Periods

The concept of critical periods in development is supported by findings related to sensory stimulation and neurological outcomes [27]. Originally identified in the visual system, critical periods are specific developmental windows during which sensory input is essential for establishing proper visual perception. If an organism is deprived of this sensory input during these critical periods, it may experience lifelong impairments in visual function [32]. Similarly, the depletion of cerebral folate during critical stages of fetal development may increase the severity of autism spectrum disorder (ASD). We postulate, following the work of LeBlanc and Fagiolini (2011) [33], that ASD may emerge from critical periods modified so that cerebral folate deficiency (CFD) adversely impacts neural development, thereby elevating the risk of a subsequent ASD diagnosis [34].

In pregnancy, the presence of folate receptor autoantibodies (FRAAs) in the mother obstructs folate delivery to the developing fetus. The FRAAs are frequently associated with pregnancies resulting in conditions such as spina bifida and ASD [35]. Identifying pregnancies with FRAA and treating expectant mothers with folinic acid or methyl-folate may facilitate adequate folate transport to the fetus, potentially reducing the risk of developmental disorders such as ASD [36,37].

To this end, Stephanyshyn [31] postulate that in women who take elevated levels of folic acid, the unmetabolized folic acid in their blood may increase the probability that their children will make FRAA. Given that folic acid is commonly prescribed for in-vitro fertilization (IVF) and other pregnancies, their finding provides a caution about use of oxidized folate (folic acid). Instead, reduced folate, in the form of methyl-folate (found in foods) or folinic acid (used in prescriptions as leucovorin), would be a safer choice for pregnancies, including IVF [38].

Additionally, diagnosing FRAA in prospective parents may offer further advantages. Recent studies indicate that supplementation with reduced forms of vitamin B-9 and B-12 can enhance pregnancy outcomes and live birth rates in women facing fertility challenges [39], underscoring the critical role these vitamins play in healthy fetal development.

Clinical research has demonstrated that in the presence of FRAA, supplementation with reduced forms of folate and vitamin B-12 can effectively counteract the blockade of folate absorption, allowing for normal brain development in children [2,26]. Folate is vital for neural health and the data show that over half of autistic individuals produce the autoantibody FRAA, which inhibits folate transport into the brain, causing CFD. Stress may exacerbate this deficiency, further complicating communication difficulties associated with ASD [39]. This stress-related impact may be mediated by elevated cortisol levels, which can reduce vitamin D—a cofactor necessary for folate transport into the brain.

Diets rich in natural folate, such as the Mediterranean diet, may help alleviate CFD, particularly when combined with strategies to reduce stress, potentially improving communication in ASD individuals.

We propose that critical periods exist in ASD development, with the first occurring in utero and a subsequent period during the early years of life (ages 1–3 years). These periods may predispose children to ASD during fetal development and set the stage for the disorder’s emergence in early childhood.

One study indicates that nutritional supplementation alongside psychological counseling is effective for children under five, [40] suggesting that therapy and nutrition should be prioritized during these formative years to maximize positive outcomes [11,41]. Additionally, preliminary reports of use of folinic acid with young autistic children who are FRAA positive indicates that treatment of those 2–3 years of age is most effective, with efficacy dropping 50% for those 4–5 years and with folinic acid being of marginal utility for those over 7 [31].

Furthermore, the association between the presence of FRAA and an increased ASD risk lends credence to the recommendation that women with FRAA who are planning to become pregnant should take prenatal supplements containing reduced forms of folate (such as methyl-folate or folinic acid). Additionally, children exhibiting FRAA or whose parents have FRAA should receive nutritional supplementation to ensure adequate levels of bioavailable vitamin B-9 for optimal brain development.

### 2.6. Diagnosis

The early diagnosis of neurodevelopmental disorders, such as autism spectrum disorder (ASD), is critical for achieving optimal treatment outcomes, including the potential for prognostic assessments during early pregnancy. One promising predictor for the development of ASD in children is the presence of folate receptor antibodies (FRAAs) in either biological parent, which can be confirmed through a blood sample [29]. Several non-invasive screening methods for ASD are currently under development, including retinal imaging, behavioral phenotype algorithms, and machine learning techniques [42,43,44].

We propose a rapid screening approach that uses family history to assess the presence of ASD, depression, or spina bifida in immediate family members of the birth parents, with subsequent FRAA testing of those screened as potential positives. Such screening systems could be implemented effectively to swiftly identify potential ASD cases. We hypothesize that testing for the FRAA biomarker in parents early in a pregnancy represents a prime opportunity to evaluate the risk for neurodevelopmental disorders, including ASD.

### 2.7. Autoimmunity and ASD

The interplay between reduced microglial activity in the central nervous system (CNS) and the effects of folate receptor autoantibodies (FRAAs) in depleting folate levels in the brain suggests a potential link to autoimmunity or immune dysfunction in the development of autism spectrum disorder (ASD). While the production of FRAAs is influenced by dietary factors, it is important to note that only a subset of individuals produce antibodies against the folate receptor. This phenomenon is similar to autoimmune conditions, where the immune system inadvertently disrupts normal physiological functions [45,46,47].

The reduction of folate in the brain has significant consequences for neurogenesis and the proper development of neural pathways. Concurrently, microglia serve as the primary immune cells in the CNS. As previously discussed, oxidative stress—stemming from infections or environmental factors—can suppress microglial activity, leading to impaired synaptic pruning. This impairment can result in an overabundance of synaptic connections, potentially affecting a child’s ability to concentrate on tasks.

This alteration in microglial function may also represent a form of CNS-specific autoimmunity. Consequently, the development of ASD may be intricately linked to immune dysregulation, characterized by systemic autoimmunity due to FRAA and stress-related immune suppression affecting microglial activity in the brain. Such immune dysregulation could have broader implications for developmental outcomes, providing insight into the common comorbidities observed in ASD individuals [39,48].

### 2.8. Molecular Mechanism of FRAA Effect

The blood–brain barrier (BBB) plays a vital role in brain health. The BBB regulates the movement of all nutrients via specific transport, using both low-affinity transporters (which move a nutrient down its concentration gradient from blood to brain and high-affinity transporters (which move a nutrient to higher concentration with an energy-dependent process).

Folate deficiency has been linked to increased risks of neurological conditions such as dementia [49]. The bioactive form of folate, L-methylfolate, can be absorbed directly in the gut without metabolic conversion and can directly cross the BBB [50]. As a crucial methyl group donor in one-carbon metabolism, L-methylfolate is integral to both the folate and methionine cycles, which are essential for DNA synthesis and methylation processes [51].

Impairments in folate metabolism hinder the conversion of homocysteine to methionine, resulting in elevated homocysteine levels that enhance intracellular oxidative stress. Furthermore, L-methylfolate supplementation has been shown to elevate levels of tetrahydrobiopterin (BH4), which is essential for nitric oxide synthase (eNOS) activity, thereby promoting vasodilation and enhancing retinal blood flow [52,53].

MTHFR polymorphisms, particularly C677T and A1298C, pose further challenges to folate metabolism, leading to the accumulation of unmetabolized folic acid (UMFA), which is associated with increased oxidative stress. Notably, these polymorphisms are common, affecting approximately 60–70% of the population, and are linked to impaired cognition and memory [54,55].

Unlike folic acid, L-methylfolate is not influenced by DHFR or MTHFR polymorphisms and has been demonstrated to be more effective in reducing homocysteine levels in the central nervous system, enhancing nitric oxide production, and providing neuroprotection [56]. In sum, L-methylfolate functions as a potent agent mitigating the impacts of oxidative stress and inflammation.

### 2.9. Treatment

For the effective management of neurodevelopmental disorders like ASD, it is beneficial if mothers with a history of FRAA receive prenatal vitamins containing reduced folate (such as methyl-folate or L-folinic acid). This supplementation should continue postpartum, with mothers maintaining folate intake while nursing. Infants should also receive folate (available in liquid form, such as from Aprofol AG) after weaning. Regular FRAA testing can guide the continuation of folate treatment during infancy and toddlerhood.

Additionally, a diet rich in plant-based sources of folate, such as legumes and leafy greens, along with reducing exposure to bovine milk products, may enhance folate absorption. Implementing these nutritional strategies can support overall health and neurodevelopmental outcomes in children at risk for ASD.

## 3. Conclusions

Autism spectrum disorder (ASD) may have its origins in utero, where critical developmental periods prime the fetus for the emergence of various neurological disorders (see Figure 1). Early identification and intervention are crucial for mitigating the severity of ASD in affected children. Promising interventions begin with nutritional supplementation aimed at restoring folate levels in the developing brain, thereby facilitating typical neurodevelopment during these critical phases. Additionally, emerging therapies hold potential for enhancing the quality of family life for those with ASD children.

There is growing evidence that critical periods in neurodevelopment may be disrupted by decreased cerebral folate levels, low taurine and/or cysteine levels, and immune activation. Cerebral folate deficiency (CFD) may play a significant role in this process. Inflammatory immune activation during fetal development could serve as a trigger for atypical developmental trajectories. Adequate folate is essential for neurogenesis, while taurine and cysteine may be vital for synaptic pruning, influencing both fetal and post-fetal neurodevelopment.

Whether the depletion of folate, taurine, or cysteine arises from maternal stress, infection, or specific genetic alleles, the implication is clear: nutritional supplementation with reduced folate, along with taurine as needed, may help restore neurotypical developmental pathways without adverse effects. We encourage the use of well-absorbed nutritional sources of reduced folate and taurine, along with testing for FRAA to evaluate CFD during pregnancy and early childhood. If tests indicate the presence of CFD, supplementation with reduced folate is recommended. In cases of infection or stress, taurine or cysteine supplementation may also be beneficial. Given that these nutrients are found in food and generally have no side effects, supplementation should pose minimal risk.

### 3.1. Figure

Taurine depletion may be addressed by reducing oxidative stress and taurine or N-acetyl cysteine (NAC) supplementation. While folate anemia may be addressed with dairy elimination and reduced folate (such as folinic acid or methyl-folate) supplementation. When both the stress and anemia paths progress, this can lead to at least one variant of ASD.

### 3.2. Table: Summary of Recommendations

With pregnancy, there are two aspects to follow regarding the potential for ASD development. These are the presence of FRAA and presence of inflammation. Table 1 looks at the recommended responses to each.

**Figure 1 cells-13-01968-f001:**
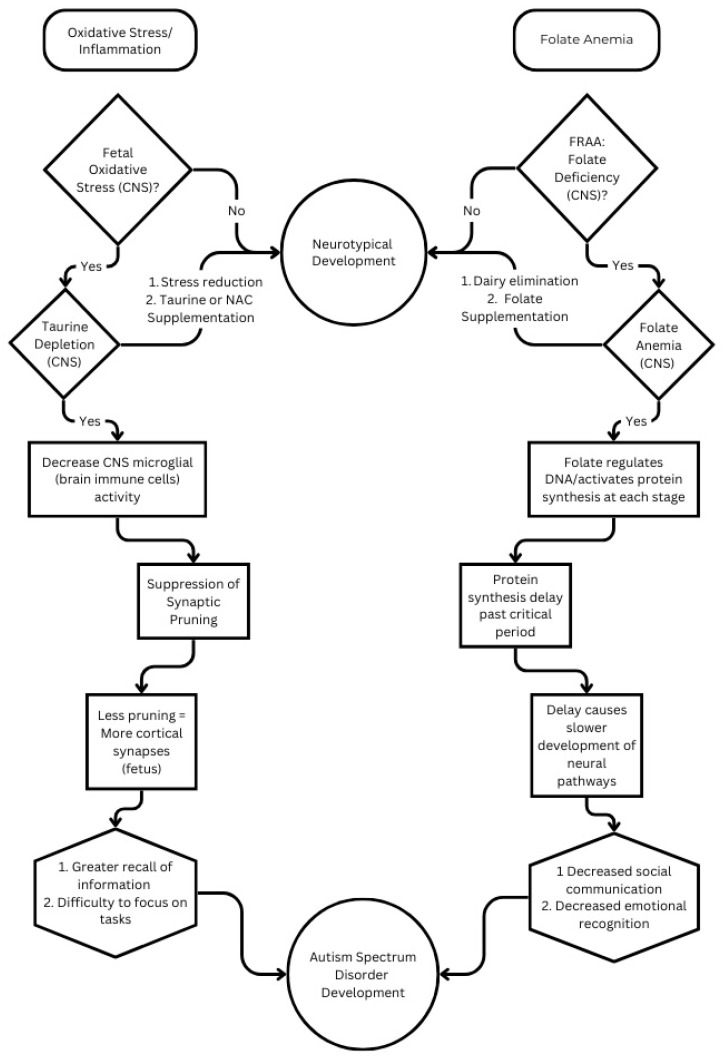
Flow chart representation of key elements in neurodevelopment.

Taurine and folate are provided by diet, processed by the microbiome, and absorbed into the blood. In neurotypical development, these two compounds support brain protection and synaptic pruning (taurine) and DNA regulation for neural pathway formation (folate). While the reduced form of folate is needed, a few hundred milligrams of folic acid (oxidized folate) can be converted to methyl folate in the gut each day. For 70% of ASD individuals, dairy triggers the production of a folate receptor antibody that prevents folate transport to the fetus or child CNS by blocking the folate receptor in the placenta or at the blood–brain barrier (BBB). Additionally, stressors elevate cortisol levels and can deplete folate or taurine. Since synaptic pruning requires microglial function, which in turn is supported by taurine, stress reduces pruning. Stress also reduces folate levels, negatively impacting development both in the fetal period and early childhood, evidencing that the impact can occur in utero by stress on the mother as well as during childhood.

## Figures and Tables

**Table 1 cells-13-01968-t001:** Summary of Recommendations.

Condition	Maternal Inflammation (Stress or Infection or Other)	FRAA Presence in Parent
Dietary/health modification(s)	Stress reduction, increase taurine-containing foods	Eliminate or reduce cow dairy products
Supplements	Taurine or N-acetyl cysteine	Methyl-folate (and try to avoid folic acid supplements)

## Data Availability

No new data were created or analyzed in this study. Data sharing is not applicable to this article.

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
