# Peer review of "Neurodevelopment of Autism: Critical Periods, Stress and Nutrition"

_cells, 2024, doi:10.3390/cells13231968_

Round 1

Reviewer 1 Report

Comments and Suggestions for Authors

Thank you for asking me to review this manuscript. The author has attempted to make a case for supplementing pregnant women and children with folate and taurine to mitigate the severity of ASD. The author argues that this supplementation is likely to have the greatest effect if supplementation is during times of major brain development. Although the hypothesis is of interest, there are components of the argument that require strengthening.

1.     The high prevalence of FRAA in ASD initially seems compelling. However, unaffected siblings and parents of children with ASD have similarly high prevalence on FRAA in some studies. In other studies, the prevalence of FRAA is lower in parents than in ASD, but nevertheless remains far higher than in the general population. It is implicit (although not explicitly stated) that the reason relates to timing of FRAA. However, there is no evidence presented that relates to timing and therefore the increased prevalence of FRAA may be some form of epiphenomenon

2.     Part of the argument for why FRAA is not an epiphenomenon relates to the positive effect of folinic acid supplementation. However, the magnitude of any effects is not stated. It is important to know whether any changes identified have real world positive impacts and in what proportion of children. In addition, I do not think that a personal communication about efficacy without any details on the study being referenced should be included as part of the argument. That section should be removed.

3.     It is not clear why FRAA in a father predisposes a fetus to later development of ASD. Some sort of mechanism should at least be posited.

4.     The role of inflammation in neurological disease, including ASD, has been extensively studied. However, modulation of inflammatory processes in ASD have not had a major clinical impact. The author should make an argument for why that is the case and why modulation of inflammation with taurine would be expected to have some sort of different effect.

Author Response

Thank you for the thoughtful analysis.

  1. True, timing of FRAA is likely the issue and the data are thin for this, with the published studies showing a somewhat weaker response for children over 15 compared to under 10, but there is little published for 0-5 years, which is when we expect the mores substantive difference.
  2. In section 2.4 I tried to address the magnitude of the response to folinic acid, and to specify that this response is seen in a majority but not all children so supplemented. I have added this explicitly in the first paragraph of 2.4.
  3. I am pleased that the reviewer also finds this curious. I have added to the 4th paragraph in section 2.4 to speculate that this may point to the genetic origins of FRAA.
  4. Excellent point. I have added to the 5th paragraph in section 2.3 to provide such an argument (that it is timing dependent) and propose a possible means to retrospectively cull from clinical data as to whether my argument is valid.

Reviewer 2 Report

Comments and Suggestions for Authors

Neurodevelopment of Autism: critical periods stress, and nutrition

Review

A brief summary

Autism Spectrum Disorder (ASD) is a developmental disability affecting over 2% of eight-year-old children globally, with critical periods in fetal and early postnatal development influencing neurotypical versus autistic outcomes. The study suggests that insufficient folate and factors like oxidative stress during these periods may increase ASD risk, and early supplementation with reduced folate and taurine could help promote healthier neurodevelopment and reduce symptoms.

General concept comments

Article:

The review is well-written and the figure in paragraph 3.1 is very informative, but it requires additional clarification. Specifically, it would be beneficial to include a table summarizing all recommendations, such as eliminating certain products or adding vitamin B-9, during the different developmental periods.

Review:

v In paragraph 2.6, please clarify and elaborate regarding a rapid screening approach.

Specific comments:

In section 2. Development 2.1 Folate , references [2] and [3] doesn't seem to be in the correct place.

Author Response

Thank you for your kind words and clear analysis

Regarding your request for clarity in the Figure: I have made a Table to go along with the Figure to summarize the recommendations from the manuscript. I hope this provides clarity without too much redundancy.

Section 2.6: I have rephrased the rapid screening proposal to explain that it could be part of a typical clinical exam, specifically the history portion, thus allowing a quick check of whether there may be value in a person being tested for presence of FRAA .

Section 2.1: I have confirmed that reference 3 is correct.  Reference 2 is no longer in Section 2.1

Reviewer 3 Report

Comments and Suggestions for Authors

This paper logically presents important factors for reducing the risk of autism. However, it lacks specific discussion on mechanisms related to ASD.

I recommend including detailed mention of specific molecular pathways reported in relation to oxidative stress and immunity, as well as the molecular mechanisms by which folate receptor autoantibodies affect neurodevelopment.

Additionally, I am curious whether there are any known long-term risks associated with prolonged supplementation of reduced folate or taurine in infants.

How does sensitivity to FRAA-related ASD vary depending on genetic variation, and how might this affect treatment strategies?

Lastly, there is a lack of supporting evidence for clinical trials and treatment, which would be beneficial if added.

Author Response

Thank you for your thoughtful analysis.

I have added a new section to discuss the molecular pathways related to stress and FRAA, as suggested. This is new section 2.8

Elevated folate supplementation has been used for decades in cancer treatment, and it is considered safe, with no reported risks. Additionally, folate in the form of folic acid has been supplemented in all grain products in the US since the 1990’s with no reported adverse effects. There is less practical history with taurine supplementation, and it is presumed harmless due to its presence in most animal food groups (meat, dairy, fish). I expect that taurine supplementation may be met with diet for most people, and with tablets for those who have a vegan diet since taurine levels are low in plants.

Section 2.4 provides the supporting clinical trial and treatment references, and I have added to it in this revision, as recommended.

Reviewer 4 Report

Comments and Suggestions for Authors

The prevalence of ASD has significantly increased in recent years, yet our understanding of its etiology remains limited, and there is currently no cure. The early onset of ASD suggests that critical periods during fetal and postnatal brain development may be disrupted. In this commentary, the author discussed the impact of folate deficiency and oxidative stress/inflammation on brain development, proposing that early supplementation with reduced folate and taurine during both fetal and postnatal stages could potentially mitigate ASD symptoms. This is a noteworthy and promising perspective in the field of ASD research.

Author Response

Thank you for your review. I appreciate your wisdom and kind words, and am honored.

Round 2

Reviewer 3 Report

Comments and Suggestions for Authors

The revisions have been successfully reflected. Thank you for your hard work.

Author Response

Thank you for your kind words, and thank you for your earlier review that helped me improve this manuscript.